# Extent of residual loss and damage from cyclones at household level: Evidence from the east coast of India

Saudamini Das 

Institute of Economic Growth, India

cyclones; loss and damage; Odisha; India; UNFCC

**Corresponding author:**
Saudamini Das;
Email: saudamini@iegindia.org

## Abstract

Efforts on loss and damage assessments primarily focus on the macro-level assessments that often overlook micro-level heterogeneity. This paper adopts a bottom-up approach by measuring household-level L&Ds from cyclones. Estimates are derived for a representative household in Odisha, India, using two case studies: a super cyclone and a very severe cyclone., Data were collected through focus group discussions and household surveys, and were valued using the then prevailing market prices. The findings suggest that the annual L&Ds for a coastal household in Odisha amount to USD 193 from a super cyclone and USD 396 from severe cyclones, measured in 2014 prices and exchange rate (1 USD = INR 60.95). While the super cyclone caused extensive losses, a substantial portion of the damage was compensated through government support and international aid. In contrast, very severe cyclones are more frequent but receive limited external assistance, leaving households to cope largely on their own. L&D assessment across different occupational groups reveals significant disparities in aid distribution and insurance coverage. Given that the area is a core zone of cyclogenesis, localised resource mobilisation and expanded insurance coverage should be prioritised, along with a fairer aid distribution mechanism, to strengthen disaster management.

## Impact statement

This study provides micro-level evidence on climate-related Loss and Damage (L&D) by quantifying household-level economic and non-economic burdens from cyclones in coastal Odisha, India – one of the world's most cyclone-prone regions. While most L&D assessments are conducted at national or sectoral scales, this research demonstrates how aggregate figures obscure the lived realities of affected communities. By combining focus group discussions with detailed household surveys, the study captures residual losses after accounting for government compensation, insurance pay-outs and private adaptation efforts. The findings reveal that a substantial proportion of losses – between 64% and 71% – remains uncompensated even after public and private responses. Importantly, less intense but more frequent cyclones impose higher annualised burdens than rare, extreme events, challenging conventional disaster financing models that prioritise high-profile catastrophes. The study also highlights distributional inequities in recovery support, with some occupational groups receiving disproportionate compensation while others bear persistent residual losses. Insurance penetration is found to be negligible, covering only a small fraction of total damages. Beyond monetised impacts, the research documents significant non-economic losses, including mental health distress, social dislocation and degradation of trust and cultural integrity, which remain largely invisible in formal assessment frameworks. By generating granular, community-level evidence, this study informs ongoing global debates on the operationalisation of the Loss and Damage Fund under the UNFCCC. It underscores the urgency of expanding inclusive risk finance, improving targeting mechanisms in post-disaster assistance, and integrating non-economic loss metrics into official assessment tools. The results are directly relevant to policymakers designing climate-resilient development strategies, international agencies allocating adaptation finance, and scholars refining L&D methodologies. Ultimately, the study strengthens the empirical foundation for equitable and evidence-based climate justice interventions in vulnerable coastal regions.

## Introduction

Climate change constitutes one of the most pressing global challenges of the twenty-first century. According to the latest Intergovernmental Panel on Climate Change (IPCC) assessments, without rapid and deep reductions in greenhouse gas emissions, the frequency and intensity of climate extremes – including heatwaves, heavy precipitation, marine heatwaves, tropical cyclones and, in some regions, agricultural and ecological droughts – are projected to increase

substantially (IPCC Summary for Policymakers, 2022; IPCC, 2022). Climate change-induced extreme events impose severe economic and non-economic impacts on vulnerable regions, frequently disrupting development trajectories and exacerbating existing inequalities.

Among climate-related hazards, tropical cyclones are particularly destructive in coastal regions, generating large-scale damage to livelihoods, infrastructure, ecosystems and human well-being. Developing countries, especially those with densely populated and ecologically fragile coastlines such as India, face disproportionately high risks owing to limited adaptive capacity, high exposure and persistent development constraints (IPCC Summary for Policymakers, 2022). Despite significant investments in adaptation and mitigation, these measures remain insufficient in many contexts. As a result, communities continue to experience residual impacts that cannot be fully avoided or addressed – referred to as climate-related *Loss and Damage* (L&D) (Boyd et al., 2017; Boyd et al., 2021).

The United Nations Framework Convention on Climate Change (UNFCCC) defines loss and damage as "the actual and/or potential manifestation of impacts associated with current climate variability and future climate change that negatively affect human and natural systems" (UNFCCC SBI, 2012). These impacts occur despite mitigation and adaptation efforts and include both economic and non-economic harms. In vulnerable developing countries, L&D reflects persistent adaptation gaps, shaped by both *soft limits* (constraints within human systems) and *hard limits* (biophysical limits of natural systems) to adaptation (Boyd et al., 2017; Sakdapolrak et al., 2023). Recognising and assessing L&D, therefore, requires careful, context-specific identification of residual impacts from climate extremes.

The issue of addressing loss and damage was first raised by the Alliance of Small Island States (AOSIS) in 1991 (INC, 1991) and has since become a central concern in international climate negotiations. The Bali Action Plan (COP-13) formally mainstreamed the L&D agenda (UNFCCC, 2007), followed by the Cancún Adaptation Framework, which initiated a work programme on L&D.

Subsequently, the Warsaw International Mechanism (WIM) for Loss and Damage was established at COP-19 in 2013 (Huq et al., 2013). Financial dimensions gained further prominence with the Glasgow Climate Pact and the announcement of a dedicated Loss and Damage Fund at COP-27. Despite these advances, debates around coverage, categorisation and assessment methodologies remain contentious (Joshi et al., 2025), underscoring the need for more empirical evidence from highly vulnerable regions.

This study contributes to the L&D literature by undertaking a community-level assessment using focus group discussions (FGDs) and household surveys in cyclone-affected coastal Odisha, India. Micro-level assessments capture how climate shocks translate into lived experiences affecting livelihoods, health, social relations and well-being – impacts that are often obscured in aggregate or national-level assessments. Such analyses provide critical insights into the scale and nature of household suffering, especially non-economic losses, and offer evidence to inform compensation mechanisms and the design of region-specific, climate-resilient development policies.

### Definition of loss and damage

Although the scope of loss and damage remains uncertain, it broadly encompasses the unavoidable impacts of climate change. While some damages can be avoided or reduced through mitigation, adaptation and climate-resilient development, others persist due to inadequate or maladaptive responses, limited awareness, infrastructural constraints or technological barriers. These unavoidable impacts constitute L&D.

Figure 1, adapted from OECD (2021), illustrates this concept. L&D is the gap (the magenta line) between the blue line, representing a hypothetical situation with no climate change, and the green solid line, representing the situation with climate change after adaptation measures have been implemented. The residual impacts captured by this gap reflect losses that cannot be fully addressed through existing interventions.

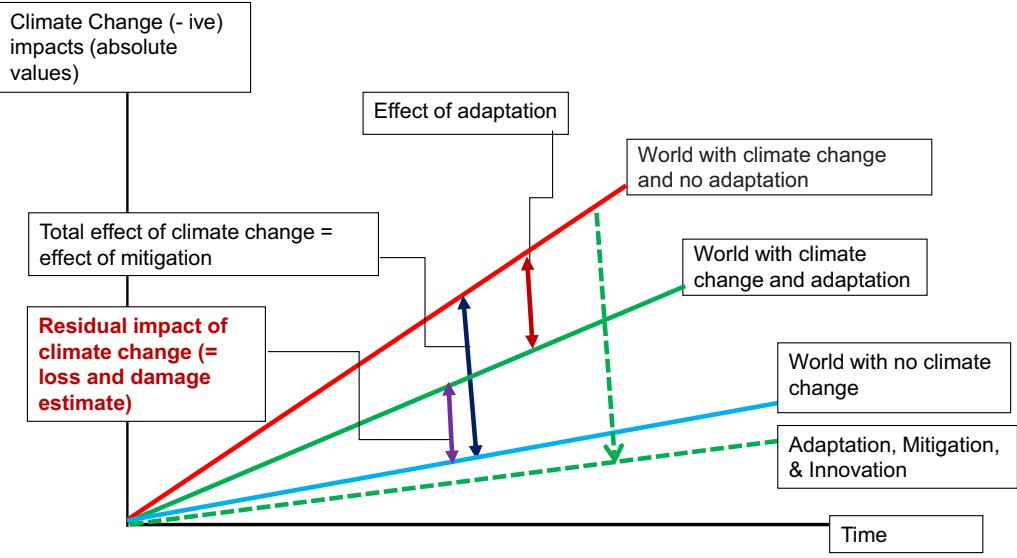

**Figure 1.** L&D as the residual impact of climate change.
*Source: OECD (2021).*

The IPCC and UNFCCC conceptually distinguish between economic and non-economic losses and between direct and indirect impacts. Direct economic losses typically involve damage to physical assets and infrastructure, whereas indirect losses arise from disruptions to production, income flows and market access. Non-economic losses include impacts on human health, mortality, cultural identity, social cohesion and ecosystem integrity – dimensions that are often difficult to monetise (UNFCCC, 2013, 2024). While economic losses can be assigned market values, non-economic losses require alternative valuation approaches, such as indirect use values or symbolic assessments.

### Identification and measurement of loss and damage

Countries such as India face heightened climate risks due to their geographic exposure, large population and development challenges (Krishnan et al., 2020). In 2024 alone, India experienced extreme weather events on 322 of 366 days, resulting in 3,238 fatalities, destruction of over 235,000 homes and damage to ~3.2 million hectares of crops (CSE, 2024). These figures highlight the magnitude of loss and damage – particularly indirect and non-economic losses – and the urgent need for systematic identification and assessment.

India has witnessed a marked intensification of cyclonic activity in recent decades in both the Bay of Bengal and the Arabian Sea (IMD, 2021). Major cyclones, such as Phailin (year 2013), Hudhud (year 2014), Fani (year 2019), Amphan (year 2020), Tauktae and Yaas (year 2021) and Remal (year 2024), illustrate the escalating burden of cyclones on coastal states. Together with Bangladesh and Myanmar, eastern India represents a global hotspot characterised by high population density, socio-economic vulnerability and increasing storm intensity (Islam et al., 2022; Siddik et al., 2024). Exposure is further amplified by rapid urbanisation, unplanned settlement in hazard-prone zones and deteriorating coastal embankments (Ghosh and Mistri, 2023).

Identifying and measuring the L&Ds is a tricky issue for multiple reasons, like the complexity of attributing the extreme events' impacts to climate change, the difficulty in quantifying both economic and non-economic losses and the lack of standardised methodologies. The non-economic damages, such as displacement, loss of life, loss of cultural heritage, mental health impacts and so on, are the most challenging, as these impacts are often subjective and difficult to quantify (Chiba et al., 2019). Although both economic and non-economic L&Ds are the outcomes of extreme events and slow-onset disasters, the latter are often not estimated, and the indicators associated with direct and indirect non-economic effects are not accounted for in the post-disaster assessment reports (Bahinipati and Gupta, 2022). However, methodological approaches for post-disaster accounting have been formalised through Post-Disaster Needs Assessments (PDNAs) and the Damage and Loss Assessment (DaLA) methodology, jointly advanced by the World Bank, UN and EU (UNDRR, 2014; World Bank, 2017). The above frameworks have been operationalised through PDNAs after major cyclones, such as Fani (Government of Odisha et al., 2019) and Phailin in Odisha (World Bank, 2013). Yet, there are gaps and critiques point to the difficulty of consistently valuing non-economic loss and damage (NELD) and integrating them into formal assessments (Bahinipati and Gupta, 2022). For effective risk management and allocation of response funds, ex-ante and ex-post L&D assessments are crucial. Financing L&Ds in terms of risk finance (to reduce unavoidable L&Ds) and curative finance (resource for unavoidable L&Ds) have

emerged as key buzzwords in climate negotiations and discourses. More empirical evidence from countries in the Global South can push the L&D debate ahead, in terms of refining the definition, concept and method.

### Magnitude of L&ds in India

In India, economic losses from cyclones have surged from US$2.99 billion in 1999 to nearly US$15 billion in 2020, with the manufacturing and industrial sectors experiencing asset and output contractions of up to 6% in hard-hit districts (Naguib et al., 2022; Saha et al., 2025). The PDNA for Cyclone Fani in the year 2019 estimated recovery needs at US$4.18 billion, with housing and energy infrastructure accounting for the largest share (Government of Odisha et al., 2019). Cyclone Amphan in 2020 devastated West Bengal and Odisha, causing massive housing losses and prolonged power outages; humanitarian assessments reported widespread infrastructure collapse (IFRC, 2021). Cyclone Tauktae in 2021 disrupted power, transport and housing across Gujarat and Maharashtra, marking it one of the costliest Arabian Sea cyclones (Government of India and Ministry of Home Affairs, 2021; IMD, 2021). Fani destroyed standing crops and fish farms in Odisha, while Amphan and Yaas inundated thousands of hectares of agricultural land with saline water (IFRC, 2021), which ripple into indirect income losses.

The Fisheries sector shows indirect losses vividly. After Cyclone Ockhi in 2017, small-scale fishers in Kerala and Tamil Nadu states of India lost income due to damaged boats, harbour closures and extended non-fishing days. Studies also noted gaps in forecast communication, which prevented effective early action (Punya et al., 2021; Martin et al., 2022). In fisheries, vulnerability is stratified by vessel size, debt levels and access to information. Small-scale fishers lacking reliable early warnings suffered greater mortality and income shocks during Ockhi (Punya et al., 2021; Martin et al., 2022). This distributional dimension is critical for designing equitable adaptation and recovery policies. Similarly, Amphan disrupted local trade networks, informal businesses and wage employment, though these losses are under-documented in monetary terms (Dasgupta et al., 2023).

Coming to non-economic L&D, research shows a heavy burden from mental health and psychosocial disorders. In the Indian Sundarbans, water insecurity and displacement after Amphan and Yaas cyclones were associated with anxiety, depression and trauma symptoms (Dasgupta et al., 2023). Ghosh and Dutta (2024) highlight intersecting vulnerabilities in health systems, showing that women, the elderly and marginalised groups experience disproportionate health risks. These findings resonate with the UNFCCC's emphasis on NELD categories such as human health and agency (UNFCCC, 2024 Ethnographic studies reveal erosion of place attachment and community identity in the Sundarbans, where repeated relocation erodes cultural practices (Sen, 2023; Tenhunen et al., 2023). Migration policy analyses further underscore that managed retreat entails loss of social networks and cultural heritage, and these dimensions are poorly captured in PDNAs (Migration Policy Institute, 2020).

Damage to ecosystems like mangrove damage, saline intrusion and biodiversity loss has been reported after recent cyclones in West Bengal and Odisha (Ghosh and Mistri, 2023). These impacts compromise ecosystem services such as coastal protection, fishery nursery grounds, carbon sequestration and many others. Despite their significance, such losses are rarely monetised in official assessments (World Bank, 2020). Village-level studies in the Sundarbans show that poorer households, women-headed households and lower-caste communities suffer higher relative losses, both economic

and non-economic (Ghosh and Mistri, 2023). Post-Tauktae needs assessments in Gujarat highlighted disproportionate housing and livelihood impacts for marginalised groups (Sphere India and Partners, 2021). Such distributional vulnerability and widening of social and economic inequality are difficult to capture in PDNA assessments.

Thus, most of the attempts to identify and assess L&Ds are partial, event-specific, sector-specific and results are with many gaps, and the community perspective and extent of net burden on the households after accounting for compensations, insurance and private adaptations are not well identified. This article analyses granular data, presents L&D estimates for households after accounting for insurance and other forms of private adaptations, provides annualised estimates and presents a comparative picture of L&D from cyclones of different intensities. The study also provides an extensive description of NELDs suffered by the community, and all these have implications for framing regional and national adaptation policies, like expanding the insurance coverage, rethinking monetary allocations for rehabilitation, relocating households from vulnerable areas and so on. The author studied households from four coastal villages affected by cyclones in coastal Odisha in India. The L&Ds assessed are due to two cyclones: one super cyclone with landfall wind velocity of 356 km/h, and another, a very severe cyclone with landfall wind velocity of 180 km/h.

## Methods

This study adopts a bottom-up, mixed-methods approach combining key informant interviews, focus group discussions (FGDs), household surveys and descriptive statistical analysis. The analysis focuses on three primary livelihood sectors that are highly sensitive to cyclonic shocks: agriculture, including livestock, marginal workers and fisheries.

Key informants from local government institutions and community organisations were consulted to identify appropriate participants for the FGDs. Two FGDs were conducted – one in Ganjam district and one in Jagatsinghpur district – to validate a preliminary list of potential loss and damage (L&D) categories identified from the literature. These discussions helped identify the specific economic and non-economic L&Ds experienced during the 1999 Super Cyclone and Cyclone Phailin (2013), the forms of public and private compensations received and the role of insurance.

FGDs were held in public buildings outside the villages and were open to nearby residents. Participation was voluntary, and all participants signed attendance registers after being informed about the purpose of the study and the confidential use of data. We promised to use the anonymous opinions without attributing anything to individuals or to villages. Each FGD was attended by 20–22 participants, including village *sarpanches* (heads of Panchayats), farmers, fishers, women, Self-Help Group members, cyclone shelter management committee members, representatives of local NGOs and elderly residents of both genders. The same discussion guide (questions) was used for both FGDs (see Supplementary Appendix). Key insights from these discussions are summarised in the Results section.

Based on the FGDs, a structured household survey questionnaire (see Supplementary Appendix) was developed to quantify the monetisable components of L&D. As the household survey had to be done in person[1] with multiple visits to villages, we decided to

[1]Many villagers were either illiterate or less educated to fill the forms themselves.

study 4 villages and 15 households with different economic background from each village, keeping their primary occupations in mind, under these logistical and budgetary constraints. Village selection was purposive, based on occupational composition and cyclone exposure, while household selection followed a stratified random sampling approach. Households were first stratified by primary occupation, and respondents were then randomly selected within each stratum. As every household in the villages was affected by the cyclone, a small number representing different economic strata was enough to provide a representative L&D figure for the region.

The questionnaire was piloted during an initial field visit and subsequently revised to improve clarity and reduce ambiguity. The final survey was administered with the assistance of a professional survey agency. Data were collected on types of damage experienced, estimated monetary values, compensation received from government and non-governmental sources, insurance pay-outs and private out-of-pocket expenditures incurred to restore livelihoods and assets.

### Study area

The study covers four coastal villages in the Indian state of Odisha: Binchannapalli, Patrapur and Puruna Bandha in Ganjam district and Trilochanpur in Jagatsinghpur district. These villages were affected by both the 1999 Super Cyclone and Cyclone Phailin in 2013. Villages in Ganjam district were severely affected by Phailin and moderately affected by the Super Cyclone, while Trilochanpur in Jagatsinghpur district experienced severe impacts during the Super Cyclone and moderate impacts during Phailin. Figure 2 shows the location of these villages in the state of Odisha.

The villages are located along the Bay of Bengal coast, a major region of cyclogenesis (Das & D'Souza, 2019). Socio-economic characteristics of these villages are broadly representative of coastal Odisha, with livelihoods dominated by primary-sector activities including agriculture, fisheries, livestock rearing and related services. Puruna Bandha is predominantly a fishing village, whereas the remaining villages have mixed occupational structures. FGDs and household surveys were conducted in 2014, a few months after Cyclone Phailin.

### Results and discussion

This section presents findings from the FGDs and household surveys. To provide a common understanding of loss and damage in the FGDs, participants were first introduced to a list of potential direct and indirect L&Ds – both economic and non-economic – compiled from the literature. These direct and indirect L&Ds, both economic and non-economic, are shown in Supplementary Appendix Table A1. Each item from Supplementary Appendix Table A1 was read out and discussed during the FGDs, and participants confirmed whether such impacts were typically experienced during cyclones. The results reported as "Yes" or "No" (Supplementary Appendix Table A1) indicate that nearly all listed forms of L&D were routinely experienced, except increased fire hazard, increased evaporation and reduction in groundwater levels, underscoring the substantial burden of cyclones on coastal communities.

Participants then described the actual L&Ds experienced during the 1999 Super Cyclone and Cyclone Phailin, particularly with respect to agriculture, fisheries, livestock, health and socio-economic conditions. These accounts, summarised in Table 1, provide a detailed picture of household suffering, some of which cannot be fully monetised or compensated.

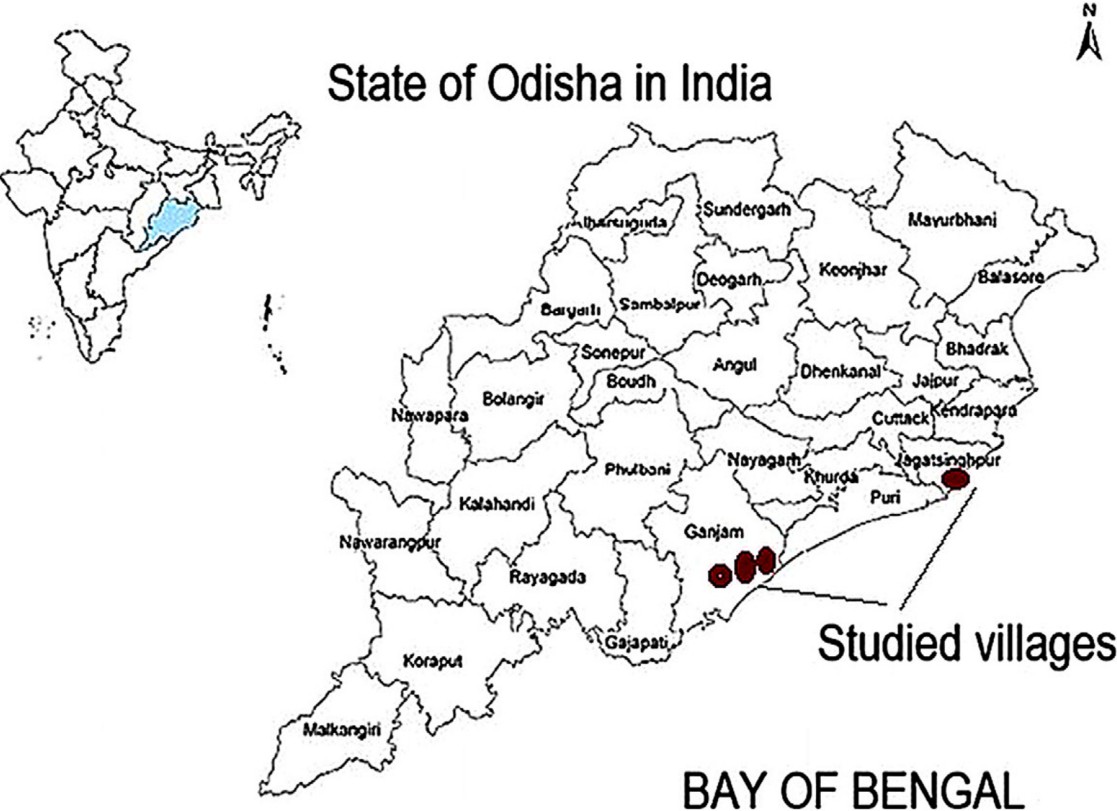

**Figure 2.** Study area villages in the state of Odisha, in Jagatsinghpur and in Ganjam district.

The household survey was designed to quantify the monetisable components of losses reported in Table 1 and to find out the net burdens. Sixty households were interviewed through face-to-face surveys conducted by trained enumerators. Respondents were primarily men aged 23–80 years, with low levels of formal education. Enumerators read the questions aloud and recorded the responses. The sample comprised farmers (37%), fishers (22%) and primary-sector service providers (41%), such as agricultural labourers and small shop owners selling primary products like fish, vegetables and so on. Approximately 40% of households held Below the Poverty Line (BPL)[2] cards. Information was collected on types of damage suffered, their approximate value, compensations received from different sources, insurance received, money spent from pocket to normalise the situation and so on. Information was collected separately for Super Cyclone and Phailin, and averaged for the entire sample and for different livelihood groups. These averaged values, with standard deviations, are shown in Supplementary Appendix Tables A2–A4. As every household did not suffer every damage and were from different economic strata, damage estimates had a wide range, with a minimum value of zero. Although recall bias – particularly for the 1999 Super Cyclone – is a limitation, this approach provides a rare approximation of grassroots-level L&D.[3]

The average monetised L&D per household was estimated at INR 273,460 (USD 4,487) for the Super Cyclone and INR 145,681 (USD 2,390) for Cyclone Phailin (Supplementary

Appendix Table A2). Losses varied across occupational groups and cyclone events. During the Super Cyclone, service providers and farmers experienced the highest losses, whereas fishers were relatively less affected (Supplementary Appendix Table A3). In contrast, during Phailin, fishers incurred the largest losses (Supplementary Appendix Table A4).

However, two- and three-sample mean comparison tests, reported in Supplementary Appendix Tables A7 and A8, did not show these group averages to be significantly different, indicating the absence of any systematic difference. Next, using the information on help received in the form of compensation and insurance, and money spent from pocket to recover the losses till the time of the survey, the uncompensated L&D and the residual L&D were estimated. Uncompensated L&D is defined as total L&D minus total compensations and insurance pay-outs received, whereas residual L&D is defined as uncompensated L&D minus private adaptations. Tables 2–4 show the estimates. After accounting for compensation received from the government and the NGOs, and insurance pay-outs, a substantial share of losses remained uncompensated. Total L&Ds on the households were USD 4,487 per household in 1999 and USD 2,390 per household in 2013 (Table 2). The high level of damage in 1999 was because of the higher intensity of the storm and the poorer socio-economic situations. It has also been seen that only 23% of the L&Ds from the super cyclone and 17% from Phailin were compensated from government and non-governmental sources. For the Super Cyclone, 77% of total L&D was uncompensated, rising to 83% for Phailin. Even after including private adaptation efforts, residual L&D constituted 64% and 71% of total losses for the Super Cyclone and Phailin, respectively (Table 2). Insurances covered a meagre 2% of the L&Ds in 1999 and 4% in 2013.

---

[2]In 2014, the BPL definition in India was a monthly per capita consumption expenditure of INR816 or less for rural areas.

[3]It is expected that people over-report losses and under-report compensations receipts, but the estimates show compensation to be higher than losses for many households for super cyclone indicating no serious recall bias.

**Table 1.** L&Ds suffered and help received by the coastal community of Odisha state in India during the super cyclone and Phailin, as disclosed from the FGDs

| Sectoral effects from the cyclones in coastal Odisha as per affected households | |
|---|---|
| Effect on agriculture | • Flooding of the village and agricultural land<br>• Loss of trees – cashew nuts, casuarinas, coconut trees, keora and other plantations<br>• Reduction in the production of salt due to the entry of fresh water into salt pans<br>• Crop loss due to brackish and barren soil from saline water inundation<br>• Loss of vegetables<br>• Loss of Rabi crops – green gram, chilli and other pulses, as most damaging cyclones come in October<br>• Less/no access to crop insurance<br>• More spending towards agricultural operations due to high input prices and out-migration of labourers |
| Effect on fishing | • Loss of fishing material – nets and boats<br>• Effect on freshwater fish due to brackish water entering the pond<br>• Much lower catch/availability for nearly 3 to 4 months after the cyclone<br>• Reduced fishing due to constant fear among fishermen in the cyclone season, as cyclones have become so frequent.<br>• Lack of financial assistance for fishing<br>• No insurance coverage for the fishery sector |
| Effect on livestock | • Loss of livestock and domestic birds – cock, hen, duck and so forth<br>• Different types of illness in livestock due to eating grasses covered with mud from flooding (the most serious issue)<br>• The arising of sanitation problems as animals and humans have to stay together in the same room/shelter for many days during cyclones, there being no shelters for livestock |
| Effect on health | • Loss of lives (deaths in some cases)<br>• Health problems during the post-cyclone period – fever, skin disease, diarrhoea, typhoid, psychological attacks and so forth<br>• Migration to other states for livelihood and poor health |
| Socio-economic impacts | • House damage<br>• Loss of utensils, cloths, seeds and all materials saved in the house, as these could not be shifted to the cyclone shelter<br>• Rise in prices of boats and nets<br>• Filling up of ponds with soil and waste products<br>• Damage to roads, community land, and community resources – school, temples, village ponds, Gram Panchayat Building and so on |
| Measures undertaken by the government | • Provided: INR 500/- and 50 kg of rice per family, Dari (tarpaulin) and a bucket to some families<br>• The 50 kg of rice could not be used properly as it got wet in the rain, flood water in many cases<br>**Other difficulties**<br>• Lack of basic facilities and special assistance for cyclone shelters<br>• There is provision of fans, overhead tanks in shelters, but they were not working due to a faulty generator and unavailability of diesel<br>• Lack of a community kitchen, drinking water facility and first aid<br>• No insurance for any type of cyclone damage, but some crop damage was compensated after assessment of crop loss |

Source: FGD conducted by the author in coastal Odisha.

**Table 2.** Estimates of household-level compensated and uncompensated L&Ds from cyclones (in INR and in prices as prevailed in the year 2014)

| Sl. No. | Type of loss, expenditure and receipts | | Super cyclone (1999) | Cyclone Phailin (2013) |
|---|---|---|---|---|
| 1 | Total monetized L&D (taken from the last row of Supplementary Appendix Table A2) | | 273,460 (USD 4,487)[a] | 145,681 (USD 2,390)[a] |
| 2 | Compensation received from the government and/or NGOs | | 57,284 | 19,095 |
| 3 | Damage covered by insurance | | 6,481 | 6,005 |
| 4 | Sum of compensation and insurance received (2 + 3) | | 63,765 | 25,100 |
| 5 | Amount of uncompensated L&Ds (1–4) | | 209,695 | 120,581 |
| 6 | Percentage of uncompensated L&Ds (5/1) | | 77% | 83% |
| 7 | Private adaptation or private replacement expenditure (money and labour time spent privately to bring back normalcy) by the time of the survey | Agricultural output | 11,083 | 3,658 |
| | | Assets and amenities | 17,250 | 11,150 |
| | | Purchase of livestock and Fishery | 5,338 | 2,360 |
| 8 | Total private adaptations made | | 33,671 | 19,868 |
| 9 | Sum of insurance, compensation and private adaptation (4 + 8) | | 97,436 | 42,168 |
| 10 | Residual L&Ds on the household (difference between 1 and 9) | | 176,024 (USD 2,888)[a] | 103,513 (USD 1,698)[a] |
| 11 | Percentage of residual L&Ds on the household (10/1) | | 64% | 71% |

[a]The US dollar exchange rate used is 1USD = INR60.95 as prevalent in 2014.

**Table 3.** Estimates of household-level compensated and uncompensated L&Ds @2014 prices (INR) for different occupational groups

| Sl. No. | Type of loss, expenditure and receipts | Farmer (*N* = 22) | Fisherman (*N* = 13) | Service provider (*N* = 25) |
|---|---|---|---|---|
| **Estimates for super cyclone (1999)** | | | | |
| 1 | Total monetized L&D (taken from the last row of Supplementary Appendix Table A2) | 307,592 (USD 5,047)[b] | 147,275 (USD 2,416)[b] | 311,155 (USD 5,105)[b] |
| 2 | Total amount of compensation, insurance and private adaptations | 49,601 | 191,059 | 67,585 |
| 3 | Residual L&Ds in the household | 257,991 | −43,784 | 243,570 |
| 4 | Percentage of residual L&Ds in the household | 84% | −30% | 78% |
| **Estimates for Phailin (2013)** | | | | |
| 6 | Total monetized L&D (taken from the last row of Supplementary Appendix Table A3) | 95,382 (USD 1,565)[b] | 235,095 (USD 3857)[b] | 143,354 (USD 2,351)[b] |
| 7 | Total amount of compensation, insurance and private adaptations | 23,898 | 88,238 | 25,106 |
| 8 | Residual L&Ds in the household | 71,484 (USD 1,173)[b] | 146,857 (USD 2,410)[b] | 118,249 (USD 1,940)[b] |
| 9 | Percentage of residual L&Ds in the household | 75% | 62% | 82% |

[b]The US dollar exchange rate used is 1USD = INR60.95 as prevalent in 2014.

After accounting for private adaptation, the residual L&Ds on the household are measured as INR176, 024 or USD 2,888 (64% of total) from Super Cyclone and INR 103,513 or USD 1,698 (71% of total) from Phailin. In comparison to the super Cyclone, cyclone Phailin was less severe, received less global attention and less resource flows, and that may have led to only 29% of the damage getting compensated, resulting in a 71% net L&D on the households.

Disaggregated analysis reveals pronounced distributional differences. Farmers and service providers consistently bore the largest residual burdens, while fishers experienced highly variable outcomes depending on the cyclone. These findings highlight inequities in compensation mechanisms and the limited effectiveness of insurance in mitigating household-level climate risks.

Table 3 shows summary L&D figures for the three groups for both the cyclones, and this table is extracted from Supplementary Appendix Tables A5 and A6, which are based on Supplementary Appendix Tables A3 and A4. From Super Cyclone, the average L&Ds on primary service providers and on farmers was around USD 5000, whereas it was much less, USD 2,416, on fishermen. Interestingly, the fishermen community managed a high amount of compensation (Supplementary Appendix Table A3) and ended up being a net gainer from the cyclone. This is also confirmed by both two and three-sample mean comparisons that show a significant difference in compensations received by fishers compared to the other two occupational groups during both the cyclones (Supplementary Appendix Table A9). The net amount gained is around 30% of their average L&Ds. The other two communities did not get enough

compensation and were burdened with a residual L&D of 84% (farmers) and 78% (service providers) of their actual L&Ds.

During Phailin, the average L&Ds of fishers was the highest, and they also managed more compensation compared to the other two groups, but still suffered a higher residual L&Ds of USD 2410, whereas it was USD 1175 for farmers and USD 1940 for service providers. The insurance played an insignificant role in L&D mitigation (Supplementary Appendix Tables A3 and A4). The fishers did not receive a single penny of compensation from insurance, and the farming community received the maximum pay-outs, followed by service providers, but the insurance amount was just 4% of the L&Ds for farmers and 1% for service providers. Disaster relief compensation from the government and private sources was the main source of help to mitigate losses for rural Odisha communities.

The L&Ds′ estimate from individual storms was annualised using the cyclone frequency over the last 30 years for the state of Odisha (see Table 4). Between 1990 and 2020, the state suffered two super cyclones and seven very severe cyclones,[4] and thus, their annual probability (the number of occurrences per year) comes out to be 0.07 for the super cyclone and 0.23 for the severe cyclones. Multiplying the residual L&Ds by these numbers, the annualised L&Ds on a household come to INR 11,735 (USD 193) for a super cyclone and INR 24,153 (USD 396) for a severe cyclone, nearly 2.06 times more, for the entire sample. For farmers, the annual L&Ds are similar for both disasters, but the annual L&D from Phailin is 1.7 times higher than that from the super cyclone for the service providers. For fishers, Phailin imposed an annual L&D of USD562, whereas they were net gainers from the super cyclone. Cyclones like Phailin are becoming almost an annual phenomenon in the Bay of Bengal region, and the poor rural communities are suffering an annual burden of around USD 400 in the form of residual L&Ds. Such a burden constitutes 60% of the current price per capita income of the state of Odisha in 2014, which means severe cyclones erode more than half of the annual household income in affected coastal areas. Less intense but more frequent cyclones, such as Phailin, impose particularly high cumulative

**Table 4.** Annual average residual L&Ds on the household in 2014 prices (in INR)[c]

| Sample groups | Super cyclone (1999) | Phailin (2013) |
|---|---|---|
| Entire sample | 11,735 (USD 193) | 24,153 (USD 396) |
| Farmers | 17,199 (USD 282) | 16,680 (USD 274) |
| Fishermen | −2,920 (-USD 48) | 34,267 (USD 562) |
| Service providers | 16,238 (USD 266) | 27,591 (USD 453) |

[c]The INR and USD exchange rates are same as shown in other tables.

---

[4]https://en.wikipedia.org/wiki/Bay_of_Bengal, accessed on 28[th] Dec 2025. The cyclone probabilities do not change if the 30-year period is taken from 1980 to 2010.

burdens while attracting relatively limited external assistance. Such types of severe cyclones are more frequent in the study region; they remain less noticed and bring in fewer resource transfers, but impose a high burden on the vulnerable poor households. Localised resource mobilisation and widening of the insurance cover need to be prioritised.

## Conclusion and significance

Assessments of loss and damage (L&D) at the macro level are common in disaster policy and aid allocation; however, micro-level analyses remain limited, despite evidence of household welfare impacts being used for disaster assessments (Markhvida et al., 2020). Household-level assessments are crucial for capturing the lived realities of vulnerability, coping capacity and recovery, and they provide essential inputs for local adaptation and mitigation strategies. Such estimates can complement the findings of macro-level Post Disaster Needs Assessments (PDNA) and provide robust evidence for policymaking. This article undertakes such an assessment by examining household-level L&Ds from two cyclones in Odisha, India: a super cyclone and a very severe cyclone. While the super cyclone resulted in higher absolute losses, the annualised household burden was little more than two times greater in the case of the less intense but more frequent very severe cyclone. These less intense events often attract limited external aid and government attention, leaving households to absorb the bulk of the costs. The other striking finding is ad-hoc-ness in aid distribution, some communities being more than compensated and some receiving very little, though all seem to have suffered similarly. Such ad-hoc-ness was observed during both the 1999 super cyclone and the 2013 cyclone, indicating some inherent flaws in the compensation mechanism. There is very low penetration of insurance and insurance pay-outs. Insurance as the damage mitigation strategy is too negligible. The findings highlight the need for enhanced local resource mobilisation, proper framing of localised aid distribution strategy and broader insurance coverage. In most developing countries, insurance penetration – particularly for climate-related disasters – remains low (Panda et al., 2020), forcing households to rely heavily on governments, NGOs and international donors. Thus, justified aid distribution in proportion to damage suffered is necessary. International aid agencies also need to reconsider allocation criteria to better address such aid gaps in regions more affected by low-intensity but more frequent severe cyclones.

The study has multiple limitations and challenges, including potential reporting biases, small sample size, purposive sampling, absence of rigorous statistical analysis, challenges in valuing non-economic losses and discrepancies between reported and compensated damages. Nonetheless, it contributes valuable ground-level evidence of cyclone-related L&Ds from a core area of cyclogenesis, especially when such micro-level estimates remain scarce globally. This study underscores the need for systematic assessments of L&Ds at the household level for fairer disaster management and resilience building.

**Open peer review.** To view the open peer review materials for this article, please visit http://doi.org/10.1017/cft.2026.10028.

**Supplementary material.** The supplementary material for this article can be found at http://doi.org/10.1017/cft.2026.10028.

**Data availability statement.** The primary data can be made available if requested

**Acknowledgements.** The data used were collected for a consultancy project sponsored by Adelphi Research, Germany, in 2014. The author thanks Adelphi Research for the financial help and the team members, especially Sibylle Kabisch, for their help. The author also thanks the community members of the study area for joining the FGD, active participation, sharing data and their experience, and providing full cooperation during household surveys. Many thanks to the survey agency, Research & Analysis Consultants (RAC), Bhubaneswar, Odisha for help in conducting the FGDs and household surveys. The author sincerely acknowledges the research assistance from Amol Amrit. The usual disclaimers apply.

**Author contribution.** The author did the household survey, collected the data, conducted FGDs, analysed the information and wrote the paper.

**Financial support.** Financial support received from Adelphi Research, under their "Climate-related Loss & Damage in India" study, is sincerely acknowledged.

**Competing interests.** The author declares none.

**Ethics statements (if appropriate).** The community members voluntarily participated in the FGD and household survey after the "purpose of the study, how the data will be used and no personal information will be made public" were made clear to them.

**Use of AI and other software.** The English grammar was cross-checked with the help of Grammarly software, and ChatGPT was used to improve the structure of the author's written sentences of only the abstract and conclusion, and no other part of the manuscript. Before submission, the author read the paragraphs carefully to cross-check the meanings of the sentences.

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
