## [Reviewer Report]

The paper addresses an important topic, but it lacks the methodological and analytical rigor required for publication. The introduction confuses background and research focus, with no clear objectives or hypotheses. The methods are insufficiently described, as the sampling design, data analysis procedures, and uncertainty handling are missing. The dataset is outdated, and the extrapolation to 2025 prices is methodologically unclear. I understand that the events used are more than 10 years old; hence, they require strict extrapolation. The results are largely descriptive, relying on self-reported values without analytical validation or comparative framing. Figures and tables are verbose but add little scientific insight. The discussion does not interpret results within a coherent framework, and overall structure and writing quality are weak. Substantial restructuring, updated data, and a transparent analytical approach would be required before the study could be reconsidered.

Recommendation: Reject due to major methodological and structural deficiencies.

---

## [Reviewer Report]

Overall review:

This work assesses the loss and damages (L&D) sustained by households from two cyclones in four villages across Odisha, India. The L&D results are obtained from a combination of focus group discussions, household surveys, and available market prices. The findings suggest that household-level analyses are crucial to consider and that household L&D risk is higher for low-intensity/high-frequency cyclones than their high-intensity/low-frequency counterparts.

Overall, the manuscript has potential to be published; however, in its current form, this reviewer recommends major revisions of this work. The reasons for this recommendation are outlined below.

Major comments:

1. It’s not clear what is novel about this work. It appears that the author is trying to distinguish macro-level L&D studies from the approach presented herein that is at the household, or a “micro-level”. After reading, I’m left wondering what new knowledge/information there is if one uses the presented approach. For example, one of the main findings – that household L&D risk is higher for low-intensity/high-frequency cyclones than high-intensity/low-frequency cyclones – could be obtained from more aggregate macro-level analyses. Furthermore, all results shown (Tables 1, 2, and 3) appear to be aggregated across the four villages thus obscuring any micro-observations that could have been made. I understand that individual privacy needs to be maintained; however, based on the abstract, impact statement, and introduction, I was expecting there to be some conclusions/findings at a more granular level.

2. The methods are not reproducible. The methods section is currently one paragraph and should be rewritten with significantly more details. This is especially important if someone wanted to expand this work to other regions or reconduct these focus groups / surveys in this same region after future cyclone events. Some example questions that came to mind when reading the methods are below.

2a. How were the focus group discussions structured?

2b. What are the questions that were asked? And how many questions?

2c. How many times was each village visited and each focus group brought together?

2d. For the survey, how many surveys were sent out and what was the response rate?

2e. For the survey, what are the questions that were asked?

2f. For the market-based valuation, where was the market data obtained from?

3. Additional details about the four villages would be very useful for readers. I’d suggest a subsection somewhere that is entirely dedicated to this, although I defer to the author’s discretion. Questions and comments regarding the villages are below.

3a. Where exactly are the four villages and Odisha located? A map showing this would help orientate readers.

3b. What is the population of each village?

3c. Are these villages small or large for this region?

3d. Do agriculture, livestock, and fisheries all play a significant role in each of the four villages? Or is one village more heavily reliant on fisheries, for example?

3e. Are the villages all equally prepared for cyclones?

4. The results presented in Table 1 and the subsequent discussion are not informative. Comments regarding Table 1 are below.

4a. Are the results in Table 1 from the focus group discussions, household surveys, or a combination of the two?

5b. How exactly are the items marked as “yes” identified as being a “yes”? For example, is a row marked as “yes” if one of the four villages identified the L&D as occurring, two of four villages, three of four, or all four?

4c. Assuming this came from the focus group discussions, did there need to be consensus within each focus group that the L&D occurred, or was an item marked as “yes” if just one participant noted that the L&D occurred?

4d. There is a note at the bottom of the table that says the list was prepared by Sibylle Kabisch and colleagues. Is there a reference for this?

4e. The text says the list in Table 1 was “verified” (page 7, line 37). How was this list verified and by whom?

4f. The current discussion of Table 1 simply concludes that there is “a heavy burden from cyclones on coastal residents”. This was previously known. I’d suggest re-creating Table 1 such that more informative information can be obtained. For example, could the responses be broken up by village and/or demographics? There may be more interesting observations to be made if there are differences across villages/demographics. This would get closer to the “micro” analysis that the author is pursuing.

5. Table 3: The results in Table 3 are where the main conclusions from this manuscript are obtained and, in the manuscripts current form, the most interesting results. Comments regarding Table 3 are below.

5a. Do the results in this table show the mean, mode, median, or some other statistic from the surveys?

5b. Similar to comment 1 and 4f, these results appear aggregated at a macro-level, rather than the desired household-level. Is there anything that can be said, while preserving individual privacy, at a finer granularity? For example, are the values the same across the four villages, do the values differ consistently across specific demographics (e.g., women tend to report X, whereas men tend to report Y), etc.?

5c. What does “Sl No.” mean?

5d. What exactly is the difference between row 6 (amount of uncompensated L&D borne by household) and 10 (Residual L&D on household)?

5e. Similarly, what exactly is meant by “private adaptations” and how does this differ from household expenditures?

5f. It’d be useful to be consistent in all rows, e.g., have USD in parentheses either in all rows or none, show percentages in all rows or none, be consistent in the use of the thousands separator, etc.

6. The conclusions are not fully supported by the results shown. The main conclusion seems to be that household-level assessments are necessary to provide a nuanced picture of L&D following cyclones; however, the results are never presented at the household, or “micro”, level. Table 3 presents what appear to be aggregated statistics of L&D, and then the manuscript ends. It’s not clear if there are outliers in the observations that significantly influence the results shown in Table 3. I’m also not sure what conclusions to draw from Tables 1 and 2 in their current form. To fully support the conclusions made and emphasize the importance of household-level analyses, additional results should be presented.

Minor comments:

1. Fig. 1: Is this figure necessary? It seems to simply repeat what is in the text.

2. Table 2: Under “Effect on Fishing”, does “3/4 months” refer to three-fourths of a month, or three to four months?

3. Table 2: This table looks incomplete. What does the last row represent? Why does the “Measures undertaken by the government” have a sub-heading “Other difficulties” in it? There’s a missing bullet in this section before “Lack of a community kitchen…”.

4. Page 12, line 17: How are the storm probabilities and annualized losses calculated? I couldn’t reproduce the numbers provided.

5. Page 12, line 36: Can the author provide a reference showing that the frequency of cyclones in this region increases? It’s my understanding that climate change generally leads to cyclone frequency decreasing and intensity increasing.

Typographical errors:

1. Page 2, line 12: “L&T” should be “L&D”.

2. Page 2, line 33: should “mall adaptation” be “maladaptation”?

3. Fig. 2: It looks like the magenta line is pointing between the green solid and green dashed lines.

4. Page 6, line 49: should this be four coastal villages and not “five”?

5. Page 11, line 53: I believe “households” should be used and not “household”.

---

## [Editor Report]

The submitted manuscript addresses an important topic - loss and damages (L&D) sustained by households from cyclones - that warrants careful scholarship. Both reviewers acknowledge this yet stress critical issues with the manuscript regarding the methodology, details of the study sites, and clarity of the results section. Both reviewers, in particular reviewer 2, give very clear suggestions about how the manuscript can be improved and resubmitted for consideration.

---

## [Reviewer Report]

I appreciate the careful consideration and response the author provided to each of my initial comments. The revised manuscript is significantly improved from the first iteration. Notable changes include an expanded methods section – with a subsection describing the study area, the inclusion of questionnaires and focus group materials in the appendix, and updated tables that disaggregate L&D by occupational groups. With these changes, I recommend this manuscript is accepted in Coastal Futures.

---

## [Reviewer Report]

The manuscript has undergone considerable revision since the previous submission. I acknowledge the substantial improvements made to the structure and the inclusion of appendix materials. However, critical methodological issues remain, preventing me from recommending acceptance at this stage.

1. Price Adjustment Inconsistency

The abstract claims household L&D of “USD 1,337 from the super cyclone and USD 2,864 from very severe cyclones (in 2025 prices).” However:

- Table 2 in the main text reports USD 4,487 for the Super Cyclone and USD 2,390 for Phailin (2014 prices)

- All appendix tables (A2-A6) explicitly show 2014 prices

- Table 4 header states “2014 prices”

- No methodology for inflation adjustment from 2014 to 2025 is provided anywhere in the manuscript

The abstract values are fundamentally different from the reported results, which raises questions about data handling. The authors must either:

- Provide a clear methodology for price adjustment and recalculate ALL values consistently throughout the manuscript, or

- Correct the abstract to reflect 2014 prices as shown in all tables

2. Missing Statistical Rigor

The manuscript remains entirely descriptive. Tables present mean values without:

- Standard deviations or standard errors

- Sample sizes for occupational subgroups (how many of the 60 households were farmers vs. fishers vs. service providers?)

- Confidence intervals

- Any statistical tests (e.g., are differences between occupational groups significant?)

- Uncertainty quantification of any kind

While I understand the dataset has limitations, the complete absence of statistical measures is problematic for a quantitative study. At a minimum, the authors should:

- Report sample sizes for each occupational category in Table 3

- Provide ranges (minimum-maximum) for key estimates

- Add explicit acknowledgment of these limitations in a dedicated limitations section

3. Inadequate Methodological Documentation

Despite the expanded methods section, critical details remain missing:

a) The largest L&D component is unexplained: In Tables A2-A4, “Amount of compensation that can restore quality of life to pre-disaster days” constitutes INR 133,750 (49% of total L&D) for the Super Cyclone. This is described only as “A subjective estimate that people gave.” What was the exact question asked? Was this a willingness-to-accept valuation? How was it elicited during surveys? This component is too large to leave unexplained.

b) Recall bias is acknowledged but not addressed: The authors mention recall bias for the 1999 event (page 11, line 4) but provide no methodological approach to minimize or account for it. Were any validation checks conducted? Were responses cross-referenced against any secondary sources?

c) Sampling justification absent: Why 15 households per village? Was any power analysis conducted? The explanation “budget did not allow for a larger sample” is insufficient for peer-reviewed research.

d) Annualization methodology unclear: Table 4 states calculations use “cyclone frequency over the last 30 years,” but the calculation is not shown. Based on the footnote (2 super cyclones and 7 very severe cyclones 1990-2020), this should be explicit in the methods or table notes.

4. Results Interpretation and Framework

The manuscript claims to provide “micro-level” or “household-level” analysis to distinguish it from macro-level studies. However:

- All presented results are aggregated means across 60 households.

- No household-level variation is shown (no distributions, no ranges, no outlier identification)

- Village-level differences are not reported despite data collection from four distinct villages

- The distinction from “macro-level” approaches is therefore unclear

What specific insights does this household survey provide that would not be captured in a Post-Disaster Needs Assessment (PDNA)? The aggregated results suggest farmers and fishers have different loss profiles, but this could be inferred from sector-level analyses.

5. Table 1 Adds Minimal Value

Table 1 occupies significant space to show that cyclone impacts occur, which is not novel. The only new information is that 3 of 41 potential L&D categories were not observed (fire hazard, evaporation, water table). This table duplicates Appendix Table A1 and should be removed from the main text or substantially revised to highlight actual findings rather than confirmations of known phenomena.

6. Define key terms in the main text: “Private adaptations” and the distinction between “uncompensated L&D” (row 5, Table 2) vs. “residual L&D” (row 10, Table 2) are clear in the appendix tables but need a brief explanation in the main manuscript.

7. Exchange rate application: The note in Table 2 states 1 USD = INR 60.95 (2014 rate). Confirm this rate was used for ALL USD conversions throughout, including the abstract.

8. Formatting inconsistencies: Thousands of separators are inconsistent between tables (some use commas, some don’t). The manuscript would benefit from careful copyediting.

---

## [Editor Report]

Thanks very much for your revised paper which addressed many of the reviewers original concerns. However, I concur with Reviewer 2 that a few outstanding issues need to be addressed prior to publication. Therefore I suggest a minor revision of the present version of the manuscript.

---

## [Editor Report]

Thanks very much for responding to the reviewers comments and improving the manuscript. I now recommend that the paper be published in Coastal Futures.